# Brain-Derived Neurotropic Factor in Neurodegenerative Disorders

**DOI:** 10.3390/biomedicines10051143

**Published:** 2022-05-16

**Authors:** Abdallah Mohammad Ibrahim, Lalita Chauhan, Aditi Bhardwaj, Anjali Sharma, Faizana Fayaz, Bhumika Kumar, Mohamed Alhashmi, Noora AlHajri, Md Sabir Alam, Faheem Hyder Pottoo

**Affiliations:** 1Department of Fundamentals of Nursing, College of Nursing, Imam Abdul Rahman Bin Faisal University, Dammam 31441, Saudi Arabia; amsudqi@iau.edu.sa; 2School of Pharmacy & Emerging Sciences, Baddi University of Emerging Sciences & Technology, Baddi 173205, India; lalitachauhan004@gmail.com; 3Department of Pharmaceutical Sciences, Manav Bharti University, Vill. Laddo, Sultanpur (Kumarhatti), Solan 173229, India; aadipharma53@gmail.com; 4Department of Pharmaceutical Chemistry, Delhi Institute of Pharmaceutical Sciences and Research, Sector-3, MB Road, Pushp Vihar, New Delhi 110017, India; anjalish092@gmail.com (A.S.); faizanazargar@gmail.com (F.F.); 5Department of Pharmaceutics, Delhi Institute of Pharmaceutical Sciences and Research, Sector-3, MB Road, Pushp Vihar, New Delhi 110017, India; bhumika201993@gmail.com; 6College of Medicine and Health Sciences, Khalifa University, Abu Dhabi P.O. Box 127788, United Arab Emirates; 100053507@ku.ac.ae; 7Department of Medicine, Sheikh Shakhbout Medical City (SSMC), Abu Dhabi P.O. Box 127788, United Arab Emirates; 8SGT College of Pharmacy, SGT University, Gurgaon 122505, India; mdsabiralam86@gmail.com; 9Department of Pharmacology, College of Clinical Pharmacy, Imam Abdul Rahman Bin Faisal University, P.O. Box 1982, Dammam 31441, Saudi Arabia

**Keywords:** brain-derived neurotrophic factor, tropomyosin receptor kinase B, neurodegenerative disorders, Alzheimer’s disease, Parkinson’s disease

## Abstract

Globally, neurodegenerative diseases cause a significant degree of disability and distress. Brain-derived neurotrophic factor (BDNF), primarily found in the brain, has a substantial role in the development and maintenance of various nerve roles and is associated with the family of neurotrophins, including neuronal growth factor (NGF), neurotrophin-3 (NT-3) and neurotrophin-4/5 (NT-4/5). BDNF has affinity with tropomyosin receptor kinase B (TrKB), which is found in the brain in large amounts and is expressed in several cells. Several studies have shown that decrease in BDNF causes an imbalance in neuronal functioning and survival. Moreover, BDNF has several important roles, such as improving synaptic plasticity and contributing to long-lasting memory formation. BDNF has been linked to the pathology of the most common neurodegenerative disorders, such as Alzheimer’s and Parkinson’s disease. This review aims to describe recent efforts to understand the connection between the level of BDNF and neurodegenerative diseases. Several studies have shown that a high level of BDNF is associated with a lower risk for developing a neurodegenerative disease.

## 1. Introduction

In the central nervous system, brain-derived neurotrophic factor (BDNF) is a significant neurotrophic factor with a major role in neuronal cell differentiation, maturation [1], and survival [2]. BDNF also has a neuroprotective effect in several pathological conditions, including cerebral ischemia, glutamatergic stimulation, decreased blood glucose, and neurotoxicity [3]. It promotes and regulates neurogenesis [4] in several regions in the central nervous system, such as the cerebral cortex, olfactory system, mesencephalon, basal forebrain, hippocampus, hypothalamus, spinal cord, and the brainstem [1]. Low levels of BDNF protein have been shown to have a role in the development of neurodegenerative disorders, such as Parkinson’s disease (PD) [5], multiple sclerosis (MS) [6], and Huntington’s disease [7]. Additionally, BDNF plays a major role in energy homeostasis along with a neuroprotective effect, and has been shown to be able to restrain calorie intake and decrease body weight via peripheral or intracerebroventricular (ICV) administration [8].

BNDF can also be found in the peripheral tissues as it can cross the blood-brain barrier; this peripheral circulating BDNF is not derived from the brain, but can be synthesized in several tissues including the liver, lung, muscles, spleen, and the vascular smooth muscles. Most of the peripheral BDNF is stored inside the platelets while the remaining amount circulates in the plasma. It has been suggested that the level of peripheral BDNF is correlated and regulated by the level of CNS BDNF [9].

According to modern estimates, neurological diseases are major contributors to disability and morbidity, and the percentage of disease occurrence is predicted to be higher in the future [10,11]. The absence of adequate therapy means that such diseases create considerable problems worldwide. The lack of adequate therapy is partially due to insufficient awareness or knowledge about the causes of most neurological disorders. Many diseases related to the central nervous system are closely connected to various environmental stimuli, such as stress [12]. Challenging events create severe risks for people with such diseases, particularly those who are more susceptible to the effects of the disease. A high level of stress sometimes has beneficial effects, such as increased attention and memory. However, stress can have harmful effects on the brain when disquieting actions become part of the daily life of an individual [13]. Various studies have shown that strain is related to metabolic problems, increased risk of heart disease, damage to endocrine functions, fluctuations in mood, and impairment of mental abilities. Stress also leads to greater risk of development of manic and neurologic disorders [13]. Persistent strain causes activation of microglia and increases the secretion of cytokines and pro-inflammatory mediators, leading to the migration of various immune cells into brain tissue, thus creating a suitable environment for the development of numerous brain diseases. In response to sustained stress, brain cells secrete several anti-inflammatory mediators, growth factors, and neurotrophic factors that can help neuronal survival [14]. BDNF has been suggested to be an important factor in various pathological conditions and is a candidate biomarker in therapies of various neuronal diseases [15]. In blood, BDNF levels significantly change in response to current treatments; therefore, it is difficult to completely understand the processes that support alteration in BDNF levels in the diseased state [15], and that reduce BDNF in the CNS and blood [16]. In summary, BDNF has a significant role in the pathology of many neurological diseases, with inflammation of neurons serving as a primary trigger for the development of brain pathologies [16,17].

## 2. BDNF Expression

BDNF is the fourth member of the neurotrophin family that consists of four structurally related factors which also include neuronal growth factors (NGF), neurotrophin 3 (NT-3) and neurotrophin 4 (NT-4). The gene that encodes BDNF includes 3′ region exons that contain the code of the pro-BDNF proteins, and, in the 5′ region, a promoter region that terminates in the 5′-exon which influences gene expression [16,17]. The gene consists of five exons with the coding region found entirely on exon V; the other four exons have the promotor region on their 5′ flanking region, and, at the 3′ end, contain a splice donor. Exon V has a splice acceptor on the 5′ region and on the 3′ end it has two polyadenylation sites. Therefore, each one of the four exons can have alternative splicing with exon V with the adenylation sites able to produce eight different transcripts of the BDNF [18]. Several experiments have shown that the 5′ region in the four exons has different responses according to various neuronal activities, as both hippocampal and cortical activities can regulate the transcripts that contain exon III [19]. It is thought that the structure of the BDNF gene is related to the stage of development, the type of tissue and the function of the BDNF protein in cellular localization. Despite the similarity of the BDNF gene across different species, there are certain structural differences that are dependent on the specific function of BDNF in that organism [20]. The gene expression of BDNF is regulated at a high level by a wide range of external and internal factors, such as the level of stress, degree of exercise and activity, brain injury, and food [21]. The level of BDNF expression was found to be very low during fetal development but to increase significantly after birth followed by another decline in expression in adulthood. BDNF is expressed throughout the entire brain with the highest levels found in hippocampal and cerebellar regions [22,23]. 

BDNF gene translation produces a proneurotrophin (pro-BDNF) [24]. Subsequently, the pro-BDNF protein is cleaved by endoproteases in the cytoplasm to produce the mature BDNF or by plasmin or matrix metalloproteinases (MMP) which occur in the extracellular matrix [25]. Then, the mature and pro-BDNF are secreted and are bound to the p75 neurotrophin receptor (p75NTR) triggering an apoptosis cascade [26]. On the opposite side, cleaved mature BDNF binds to the tyrosine kinase B (TrkB) receptor which is of higher affinity. Binding to the receptor triggers various signaling cascades that include the Ras-mitogen-activated protein kinase (MAPK), the phosphatidylinositol-3-kinase (PI3K), and the phospholipase Cγ (PLC-γ) pathway [27]. These pathways increase the influx of Ca^2+^ which leads to activation of transcription factors by phosphorylation and increased BDNF gene expression (Figure 1) [27]. 

Although pro-BDNF can activate the apoptosis pathway, it is not known if it is secreted by neuronal cells under normal circumstances because the concentration of the pro-BDNF is less than that of mature BDNF [10]. This has been demonstrated in animal research which showed that the mature BDNF concentration was ten times more than the concentration of pro-BDNF [28]. This raises the issue of whether pro-BDNF is an effective factor for signaling. BDNF function is widespread in multiple regions within the brain [21]. The function of BDNF includes participation in neuronal plasticity, the survival of neuronal cells, the synthesis of new synapses, the branching of dendritic cells, and the adjustment of neurotransmitter activity between excitation and inhibition [29]. The activity of BDNF is seen during all developmental stages and within different age groups [30].

These observations suggest that BDNF and pro-BDNF have opposite biological activity; thus, post-transitional control and how pro-BDNF is processed can have an important influence on the biological activity of BDNF [31]. A significant change in the BDNF function could be achieved by the process of adding a pro-domain to the gene of BDNF. This pro-domain has a role in the folding of the BDNF protein [32]. Genetic polymorphism in the pro-domain that arises from a valine to methionine substitution in the 66 codons (Val66Met) results in a change in memory function and affects the BDNF secretion process [33]. This suggests that the pro-domain region is the area which contains most of the function in the gene for BDNF [31]. In transgenic mice where the mutation Val66Met was induced, altered anxiety behavior was expressed [34] as well as alteration in NMDR-dependent neuronal plasticity in the hippocampus region [35]. Experiments performed on hippocampal slices from transgenic mice showed that if mice were injected with BDNF pro-peptide it inhibited LTD in the hippocampus [36]. BDNF’s role in transmission at synapses requires further investigation—many pieces of research suggest that it has a significant role in enhancing the efficacy of synaptic transmission in the hippocampus and the cortex [37,38]. Supporting evidence has included the administration of K252a, which is a TrK receptor inhibitor, or TrKB-IgG, which resulted in the prevention of the induction of long-term potentiation in these areas [39]. 

## 3. Pathological Mechanism of Action

BDNF and NT-4/5 can bind to the TrkB receptor, in contrast to NGF, which can bind to several receptors, such as Trk A, C subtypes, and NT-3 [40]. TrkB occurs in two similar forms: a full-length type, abbreviated to Gp145 TrkB, with a molecular weight of 145 kDa, and a truncated form, abbreviated as Gp95 TrkB, with a molecular weight of 95 kDa. The truncated form differs from the full length form in that it lacks the tyrosine kinase domain and shows lower affinity to the nerve growth factor receptor (LNGFR); it is denoted p75 NTR [41]. LNGFR is involved in processes that are pro- and anti-trophic, such as neuron growth and death. BDNF and gp145TrkB have broad expression in brain cells; BDNF receptors are also found in the spinal cord, specifically the gray matter neurons [42].

### 3.1. Activation of TrkB

The signaling of neurotrophin has a significant role in maintaining the survival and proliferation of cells, the reduction of neural precursors, and axon and dendrite growth via TrkB receptors [43]. The NTRK2 gene encodes for neurotrophic tyrosine kinase in human beings [43,44]. TrkB has extracellular domains with multiple glycosylation sites and a transmembrane area with an intracellular domain that has Trk activity. Once activated, G proteins, such as Ras and MAP kinase, regulate the PI3-kinase and phospholipase-C-γ (PLC-γ) pathways [45]. Signal activation is faster than deactivation; activation needs two minutes, while the deactivation requires around thirty minutes in the spinal cord. Trk signals are regulated by various mediators [40,43]. Nonetheless, other small G protein messengers have a significant role in BDNF signaling, such as Ras, Rap-1, and Cdc-42-Ra [2].

### 3.2. Secondary Messengers Activation 

The Trk receptor family (which includes TrkA-C) and LNGFR regulate the function of the different types of neurotrophins. The pre-synaptic p75 NTR plays a role in regulating the binding to the Trk receptor, ERK activation by Ras, neurite protuberance and activation of terminal kinase (JNK), which causes apoptosis in different neuronal cells [2]. BDNF signaling activates various secondary messengers in the spinal cord, including via ERK signaling, the proto-oncogene c-fos, and neurons that produce nitric oxide [2,46].

### 3.3. Signaling Cascade in BDNF

Tyrosine residue activation by BDNF leads to the stimulation of pathways that affect neural plasticity, neurogenesis, cell survival, and resistance to stress, which shows the pro-survival function of Trk receptors. BDNF signaling results in the activation of several transcription factors, such as CREB and the CREB-binding protein (CBP), which modulate gene expression that encodes several proteins which participate in different neuronal functions, such as the plasticity of neurons, neuron response to stress, and survival of neurons [43,47,48].

### 3.4. Ras/MAPK/ERK Pathway

Activation of the TrkB receptor by BDNF leads to dimerization of the receptor and phosphorylation of tyrosine residues; this creates a site for the src-homology domain which contains the Shc adaptor protein and phospholipase C(PLC). The Shc binds to both the receptor and the protein Grb2 through the nucleotide releasing factor SOS. This leads to activation of RAS [49].

Ras activation stimulates the Ras/MAPK-ERK pathway, PI3-K pathway, and PLC pathway. MAPK/ERK is essential for the formation and survival of neurons by activating several genes responsible for the survival of the neurons and inhibition of apoptosis [50]. Research concerning immature neuronal cells showed that activation of the RAS protein via BDNF protected the neurons from MK801-induced apoptosis [51]. In diseases such as schizophrenia, very low levels of ERK signaling proteins in the prefrontal cortex have been observed [52]. 

### 3.5. IRS-1/PI3K/AKT Pathway

Other pathways that participate in the actions of BDNF are insulin receptor activation, substrate-1 (IRS-1/2), PI-3K, and protein kinase B (Akt). Ras stops apoptosis by PI3K that activates pkB via deposition of the protein involved in the apoptosis pathway away from their targets [50]. Thus, the Ras-PI3K-Akt pathway is crucial for the survival of neuronal cells, and any deactivation of this pathway will reduce neuronal survival [53]. BDNF can protect hippocampus neurons from the effects of glutamate and norepinephrine via this pathway. In the hippocampus, BDNF autocrine loops result in low stimulation of NMDA receptors which means low glutamate effects and excitotoxicity [54]. Evidence from postmortem studies indicated the relationship between alteration in the Akt and Erk signaling pathways and schizophrenia; there was a low level of AKT1 mRNA proteins in the cortex and the hippocampus [55], and there were some genetic variants of the AKT1 gene0020 that have been linked to schizophrenia [56]. 

### 3.6. PLC/DAG/IP3 Pathway

The binding of BDNF to the Trk receptor initiates phosphorylation of the PLC-γ protein that leads to membrane lipids being broken down into inositol 1,4,5 triphosphate (IP3) and diacylglycerol (DAG) [57], with the former inducing calcium influx and later activating protein kinase C which is needed for neurite outgrowth [58,59].

## 4. Functions of BDNF

BDNF has generally been recognized in humans as a protein compressed from the BDNF gene. BDNF was initially isolated from the pig [58]. BDNF is the foremost neurotrophic factor that has been discovered [60]. It acts via the protein tyrosine kinase receptor (TrkB) [61] and is generally associated with aspects of nerve growth [49].

BDNF plays a crucial part in the process of neuro-regeneration [62,63] by preventing neuronal death particularly in the peripheral nervous system [64]. It helps to promote the growth of immature neurons and increases the efficiency of adult neurons [65]. Inside the brain, the role of BDNF is essential for the mediation of synaptic function and the morphology of the neurons rather than being a survival factor [66,67]. Furthermore, BDNF plays a significant part in memory function as it has a role in the formation of memory [29], synaptic plasticity [68], synapse formation [69], synaptic efficacy and neuronal connectivity [70]. In the striatum, the loss of BDNF signalling results in spinal atrophy, which is caused by a defect in the GABAergic spiny neuron in the striatum [71]; striatum GABAergic neurons do not produce BDNF, but they obtain it from the presynaptic neurons of the cortico-striatal projections [30]. Disruption in the axonal BDNF transport to the striatum from reduced cortical supply results in the degeneration of striatal neurons, a pathological feature of Huntington’s disease [72]. BDNF in the periphery exists in the plasma, platelets, and the serum [73]. The formation of BDNF is generally performed by vascular endothelial cells and secondary blood mononuclear cells [74]. Observations of polymeric markers has demonstrated an association of bipolar disorder with BDNF in large samples, particularly the Neves-Pereira samples [75,76]. 

Animal studies have shown that neurotrophins are found in high concentrations in the hippocampus and the hypothalamus, suggesting a significant role of BDNF in learning and memory [21,77]. The decline observed during aging involves multiple factors that influence several systems. In the case of learning and memory processes which are severely reduced with aging, it has been found that these cognitive effects result from impaired neuronal plasticity, which is altered in normal aging but particularly so in Alzheimer’s disease. Neurotrophins and their receptors, notably BDNF, are expressed in brain areas exhibiting a high degree of plasticity (i.e., the hippocampus, cerebral cortex) and are considered as molecular mediators of functional and morphological synaptic plasticity. The modification of BDNF and/or the expression of its receptors (TrkB.FL, TrkB.T1 and TrkB.T2) have been described during normal aging and in Alzheimer’s disease. Interestingly, recent findings have shown that some physiological or pathologic age-associated changes in the central nervous system could be offset by administration of exogenous BDNF and/or by stimulating its receptor expression. These molecules may thus represent a physiological reserve which could determine physiological or pathological aging. These data suggest that boosting the expression or activity of these endogenous protective systems may be a promising therapeutic alternative to enhance healthy aging [21,78,79]. With aging, reduction in neuronal plasticity in the hippocampus and the hypothalamus leads to impairment in learning and memory function [80]. BDNF participates in synaptic plasticity and can protect the neuron from several brain insults [81]. A systematic review and meta-analysis conducted in 2019 found that patients with Alzheimer’s disease have lower levels of BDNF, especially during the late stages of the disease [82]. Furthermore, BDNF plays an important role in learning and memory formation; El Hayek et al. found that during exercise in male mice, lactate metabolite enhanced hippocampal-dependent learning via the BDNF pathway [83].

Administration of a single dose of BDNF into the hippocampus resulted in improvement in memory and emotional behaviour in rats [84], while chronic administration of BDNF was shown to have positive potentiation effect in neurotransmission in the hippocampus region [85]. Moreover, a strong connection between the level of BDNF mRNA in the hippocampus and the memory function has been reported in rats [86]. In the hypothalamus, BDNF can participate in a neurohormonal role via the induction of synthesis, and the release of these hormones [87,88], while the expression of BDNF varies in response to different physiological stimuli [89].

### 4.1. BDNF and Aging

Aging is a process that consists of multiple declines in endocrine, cognitive, and immunological functions. Several factors play a major role in determining the aging process and the outcome, such as genetic, epigenetic, and environmental factors [90]. In most cases, there is a decline in cognitive capabilities linked to altered hippocampal and cortical functions [91]. Moreover, memory is affected significantly by the aging process, though the degree of memory impairment varies between individuals and the types of memory involved [92]. Generally, effective cognitive function is associated with optimal neural plasticity that is markedly decreased with aging [93]. Change in learning function is not usually associated with neuron loss [94]. The cognitive and learning function changes have been correlated to decreased BDNF expression and signaling [94]. There is impairment in BDNF-induced LTP due to changes in receptor function, which has been attributed to age-related effects.

The administration of Ampakine can induce expression of BDNF, which can revert changes in the neuronal plasticity function, as observed by some researchers [95]. Ampakine modulates AMPA receptors that can restore LTP to the basal dendrites, which is shown to improve memory function in rats [96]. Certain environmental influences can reverse the decline in hippocampal plasticity and reduced neurogenesis in the dentate gyrus, which are also linked to a high level of BDNF in the brain [97]. Numerous studies that have been performed on humans and animals have confirmed that in the aging brain there is a significant decline in BDNF and TrkB receptor expression [98,99,100], while spatial learning tasks have been shown to reverse or normalize receptor levels, as seen in aged Wistar rats [101]. BDNF system activation has been shown to enhance a healthy aging process, and the administration of exogenous BDNF may be able to regenerate neurons in certain neurodegenerative diseases [21].

### 4.2. The Role of BDNF and Alzheimer’s Disease

There is growing evidence of a relationship between a decreased level of BDNF expression and AD [102,103]. The pathological characterization of AD is associated with the accumulation of β-amyloid peptides (Aβ) in the brain with an increased level of hyperphosphorylated cleaved tau microtubules [104]. The impaired metabolism of the β-amyloid peptides results in neuritic plaque (NP) formation, where the hyperphosphorylated tau causes neurofibrillary tangle (NFT) formation. These events result in neuronal degeneration causing dementia [105]. Several studies have provided evidence that BDNF/TrkB signaling has an essential function in amyloid processing [105,106]. This suggests that BNDF has an important role in LTP and dendritic development, which are crucial elements in memory function, by supporting synaptic integrity through the modulation of the glutamate receptors, AMPA, and NMDA [107]. In neuronal cell culture, BDNF can reduce Aβ amyloid production [95], while in the absence of BDNF, it is elevated [108]. Studies on animal models have shown an increased level of truncated TrkB receptors in the cortex of mice with AD, which has further worsening effects on spatial memory; moreover, overexpressed truncated TrkB receptors disrupted BDNF/TrkB signaling in AD [109]. NFT and NP accumulation within the hippocampus has been shown to be associated with dysregulation of BDNF and TrkB [110]. The BDNF Val66Met polymorphism has been found to be associated with profound memory decline, particularly in the preclinical phase of the disease [111]. According to the above observations, BDNF has protective effects against AD [112]. Thus, studies on AD models suggest the delivery of the BDNF gene as a possible therapeutic option for individuals with AD [113]. Additionally, Wang et al. identified the link between exercise and BDNF levels as a therapeutic method for patients with AD; it was shown that exercise induces the expression of BDNF, especially within the hippocampus, which facilitates memory and cognitive function [114]. Table 1 summarizes some research that has addressed the relation between BDNF and AD.

### 4.3. The Role of BDNF in Parkinson’s Disease 

PD is a neurodegenerative disease with motor and non-motor manifestation. The main pathology is the degeneration of dopaminergic neurons in the substantia nigra (SN) [120]. Studies have suggested that high expression of BDNF in the SN can maintain the survival and differentiation of the dopaminergic neurons [121]. Studies on PD animal models showed that infusion of BDNF can recover the destruction of dopaminergic neurons and D3 receptors [121,122]. The knockout mice did not experience BDNF reduction in the dopaminergic neurons and the D3 receptor [121]. BDNF had a protective effect on hippocampus neurons from oxidative damage that resulted from injury or inflammation that resulted from the induction of heme oxygenase; the protective mechanism was modulated via RAS-MAPK and the P13K-AKT signals, which induced Nrf2 nuclear translocation [123]. Endoplasmic reticulum (ER) stress can induce apoptosis of dopaminergic neurons causing PD [124]. ER stress caused apoptosis by glycogen synthase kinase 3β (GSK3) activation, cyclin D1 suppression, and AKT inactivation. TrkB overexpression stopped these mechanisms via activation of the AKT signal pathway, causing overexpression of cyclin D1 and enhancing the phosphorylation of GSK3, with the net effect of preventing apoptosis of neuronal cells [125]. Furthermore, α-Synuclein (α-syn) mutations were associated with reduced TrkB and BDNF [126,127]. Some studies have found that BDNF/Trkb axonal signaling transport was significantly reduced in axons with an α-syn aggregate [128]. α-syn can interact with the TrkB receptor, specifically the kinase domain; this interaction induces its ubiquitination, reducing TrkB expression [129], as shown in Figure 2. The presence of BDNF can inhibit this interaction, thus blocking the destruction of it [130]. One recent study has linked the BDNF genotype and response to long-term pharmacological therapy in Parkinson’s disease patients, suggesting a role of BDNF in prediction and counseling in the treatment of Parkinson’s disease [131]. Further studies should be implemented to assess the possible therapeutic effect of BDNF in PD (Table 2).

### 4.4. Potential Biological Impact of BDNF Markers 

According to a study undertaken by Weinstein et al. relating to the connection between the level of BDNF in the serum and the risk for developing dementia, a ten-year follow up of the effects of BDNF in a particular community showed that, out of 2131 participants, 140 participants had a positive risk of dementia and 117 of them were at positive risk of Alzheimer’s disease [125]. Another study found that control participants with higher serum BDNF levels were at minimum risk of developing dementia and AD. Interestingly, this vital relationship of serum BDNF with threat of incidental dementia and AD was restricted to women, participants aged >80 years, and those with a college degree [125]. Further studies suggested that older women were at the highest risk of AD, as they have low serum BDNF, which plays a role in AD growth. Serum BDNF might also operate like a new interpreter in healthy adults [125].

Enzyme-linked immunosorbent assay kits are also known as BDNF antibody [139] of pro-BDNF, which was recognized by Weinstein et al. After use of these kits, there was ease in detecting BDNF forms in humans; it was reported that after using the kits much higher levels of pro-BDNF and mature BDNF were observed in the serum [139].

In a cohort study, given the higher levels of both forms in human serum plus a presumed divergent purpose, it is necessary to measure the individual serum level of both forms (pro-BDNF and mature BDNF) in a technical way [140]. In addition, precautionary drugs, mainly 7,8-dihydroxyflavone (active TrkB), are being used for dementia. These therapeutic drugs enhance recorded low serum BDNF in healthy individuals, and for people who later are likely to developing dementia or AD [140]. The main factors affecting the circulation of BDNF levels in the body are caloric constraints and other physical activities. BDNF acts as an intermediate among the observed links between the threat of dementia and lifestyle [141]. Tracking of original and young applicants was maintained from the year 1992 and the year 1998 for almost ten years in the Framingham study. The study applied Cox models to relate the threat of dementia and AD with levels of BDNF adjusting for probable or potential confounders [125]. 

BDNF is one of the main modulators of AD risk. Ecological features associated with variation in BDNF brain levels, such as physical activities, might generate neuronal vulnerability and affect risk of AD [114]. Trials (excluding trials on humans) involving the release of BDNF to the brain is a dynamic and new area in translational research for AD. A further study revealed that risk of AD is connected with BDNF blood levels [125]. This study encouraged FHS researchers to explore in more detail the detailed association between BDNF in serum with the threat of dementia and AD [125]. 

The conclusion reported by Weinstein et al., 2014, in JAMA Neurology was that there was an association between risk of dementia, serum BDNF, and AD in non-demented people who were followed up for up to 10 years [125]. Higher BDNF levels were most frequently related to lesser risk of the disease. Analysis by subgroup suggested the association was restricted mainly to college-going people, women, and in the age group of people over 80 years. Increasing serum BDNF levels led to fewer consequences when account was made for homocysteine levels and other substantial activities affecting AD risk. 

The suggestions provided by these perplexing associations are not satisfying and are not easily understood; a more detailed explanation regarding these crucial findings should be less complicated and easier to understand. Certainly, ten years after diagnosis for dementia, levels of serum BDNF were found that were initially observed at the treatment stages of Alzheimer’s disease. The symptoms are expressed in terms of slight transformations in primitive functions and biased concerns. Thus, the FHS data showed that BDNF levels act as remedial indicators, highlighting the connection of sequential neurobiology rather than risk modulation of AD with BDNF. Increasingly, work has been conducted to study the BDNF genes and the active process of BDNF delivery to the brain; moreover, detailed studies regarding physical exercise and its involvement in BDNF have been pursued. The risks of reducing homocysteine or inflammation (Lyketsoset al., 2007) were shown to be modulated by the number of therapeutic trials by introducing a blood indicator (collected from the periphery) of AD risk; although the results computed were unsatisfactory. Work on the measurement of BDNF should be sustained as there is strong evidence of its significance. 

The study results regarding BDNF levels are quite interesting, but the explanation for these is complex due to various concerns. The main question studied was the connection of serum BDNF with brain levels. Levels of BDNF are affected by additional factors that are considered to influence AD risk, such as physical activity [142] and caloric restriction. Serum BDNF is a relative meandering pointer of these ecological issues rather than a direct modulator of risk. The data collected indicates that the role of BDNF is not clearcut in the mechanism of AD but shows that serum BDNF is strongly associated, being more reactive in the case of dementia. Consideration of peripheral BDNF as a working indicator for the measurement of the effects of remedial interventions is quite premature, and additional studies are necessary to determine the therapeutic and diagnostic importance of the data [143].

One recent meta-analysis study has reported a correlation of BDNF levels with acute stroke; it was concluded that the level of serum BDNF is significantly lower in stroke patients compared to controls. At the same time, there was no correlation between BDNF level and the degree of the infarct or the outcome for the patient [144].

## 5. Recent Advancements and Challenges

The main challenge in the use of BDNF as a therapeutic intervention in neurological disorders is its ability to reach the CNS in the desired concentration to elicit a therapeutic response [145].

BDNF is a polar protein that is moderate in size; according to these characteristics, it will not cross the blood-brain barrier when administered peripherally. Therefore, it needs to be administered directly into the brain [1]. 

The administration of BDNF in the ventricles of the brain or intrathecally into the CSF does not result in sufficient penetration to the brain parenchyma [146]. However, the concentration of BDNF is improved when the BDNF is conjugated to polyethylene glycol (‘pegylation’) as shown in rats, even though this may not produce a sufficient concentration to be suitable for treatment in humans [147].

Moreover, BDNF administration is associated with side-effects with long-term use that cannot be tolerated, including weight loss, dysesthesia, and Schwan cell migration into the subpial space; these side effects result from the penetration of BDNF to the superficial layers [148]. Ideally, BDNF should be administered in a system where it could achieve enough therapeutic concentration in degenerated neurons with limited distribution to other neurons to decrease the side-effects. In addition, the concentration should be maintained for a sufficient time [145]. There are several methods that could be used to enable BDNF as a therapy. These include the following: 

BDNF protein infusion which involves the direct intraparenchymal administration of BDNF [149,150]. Several clinical trials performed on Parkinson’s disease involved use of implanted devices to infuse BDNF, with high flow rates needed to achieve the desired concentration in the putamen. The method was associated with tissue damage, as evidenced by MRI. At the same time, the lower rates were not sufficient to elicit a therapeutic response [150,151]. Another observation was that the flowback of the protein in the needle resulted in distribution of the BDNF into the CSF, which caused neuronal death, as shown in animal studies. Therefore, this method needs further improvement to be suitable, such as using an infusion system that can stop the reflux of the protein and a catheter that can distribute the BDNF uniformly into the desired brain regions [152].

The gene delivery method is thought to be a safe and effective way to deliver BDNF into the desired tissue and to decrease the spread to other tissues [153]. This method has been tested in the treatment of Alzheimer’s disease, using adenovirus as a viral vector to deliver BDNF gene into the nucleus basalis [153]. It has also been tested in the treatment of Parkinson’s disease [154]. 

Other methods include the use of compounds that can stimulate the synthesis and secretion of BDNF [155]. Moreover, a different approach has been proposed via the enhancement of Trkb receptor activation by agonists such as 7,8-dihydroxyflavone [156], a compound that mimics BDNF such as LMA22A-4 [157], transactivation of Trkb, and, finally, use of facilitators of receptor effects, such as adenosine A2A receptor agonists [158], bearing in mind that the effectiveness of these methods could be altered by the receptor endocytosis effect [159]. 

## 6. Conclusions

BDNF is one of the neurotrophic factors that modulate its function through the TrkB receptor. It plays an important role in the central nervous system by the formation and maintenance of a healthy neuronal environment most prominently reflected in cognitive and memory function. Decreased activity of BDNF has been associated with the aging process and with neurodegenerative disorders. The role of BDNF in treatment and as a biomarker for diseases should be investigated thoroughly in future research.

## Figures and Tables

**Figure 1 biomedicines-10-01143-f001:**
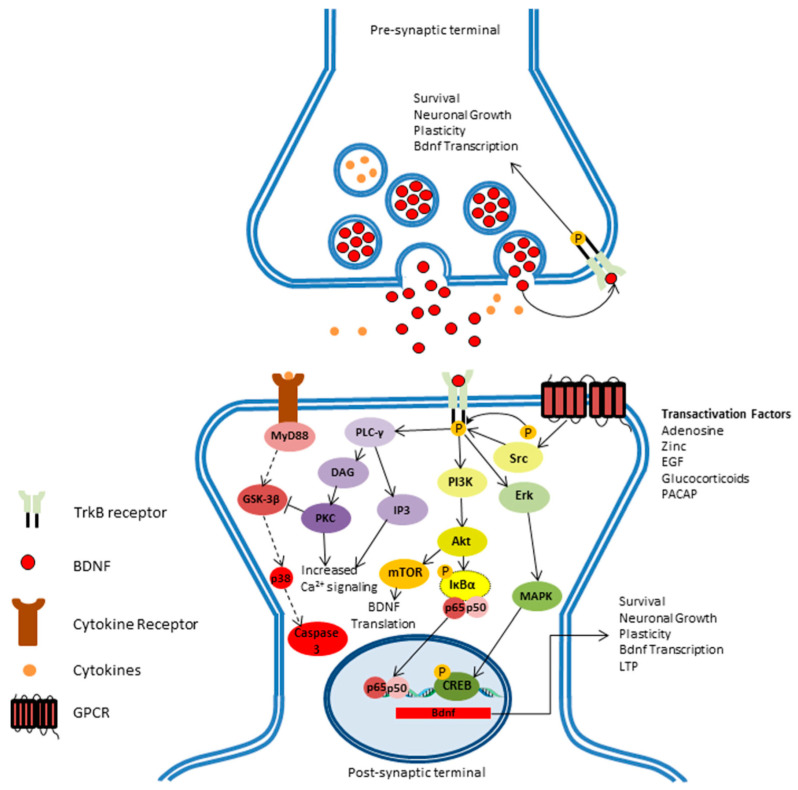
The figure shows the pathways by which BDNF signals can promote the survival of neuronal cells. The binding of BDNF to TrkB receptor switches on 3 different signaling pathways: the first pathway is the activation of the (PLC-γ) pathway which increases the level of Ca^2+^ that will terminate the apoptosis that is caused by inflammatory mediators (dashed lines), achieved by inhibiting the glycogen synthase kinase 3-beta (GSK-3β). The second pathway is activation of mTOR-dependent translation through the (PI3K) pathway, resulting in the transcription of BDNF mRNA. Additionally, the induction of Akt and Erk downstream enhances gene regulation through the NF-κB and CREB transcription factors. The third pathway is regulated by several factors such as zinc, epidermal growth factor, glucocorticoids, and the so called neurotrophic pathway, which is considered to be independent for BDNF as it can transactivate the TrkB and has a role in its signaling. This figure is adapted after modification from open access [10].

**Figure 2 biomedicines-10-01143-f002:**
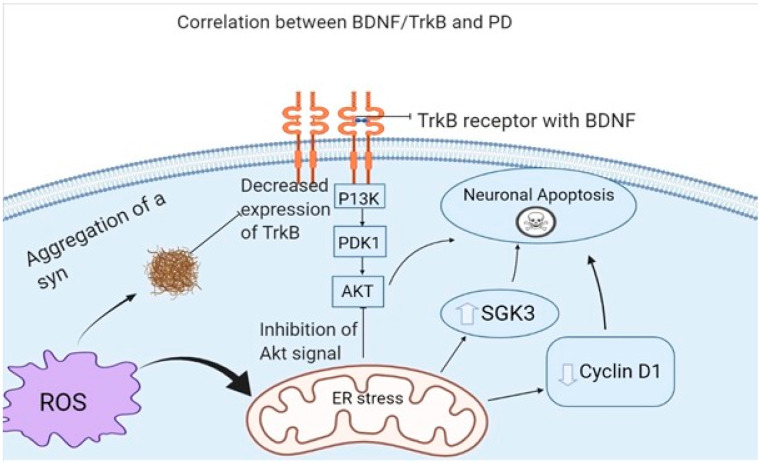
This figure is adapted after modification from open access [132]. The figure shows that ER stress will induce neuronal apoptosis by suppression of cyclin D1, activation of GSK3, and inhibition of Akt signaling from the BDNF/TrkB. A syn aggregation can decrease the expression of the TrkB receptor which will further result in loss of neurons.

**Table 1 biomedicines-10-01143-t001:** Levels and effects of BDNF in Alzheimer’s disease.

Study Objective	Sample Origin	BDNF Status	Assay Used	Conclusion	Ref.
**To determine the stage in which BDNF reduced**	Postmortem cortex	Declined	Western plot	The early stages were associated with decreased BDNF.	[103]
**A meta-analysis to examine serum BDNF in patients with AD and mild cognitive impairment (MCI) compared to healthy controls**	Peripheral serum of BDNF	Declined	NA	A systematic review and meta-analysis, comprising 15 studies, suggested that a significant decline in peripheral BDNF can only be detected in the late stages of Alzheimer’s disease.	[82]
**To explain the selective vulnerability of certain neurons to AD**	Postmortem cortex	Decreased	Western plot	Reduced BDNF may have a role in the selectivity in neuronal degeneration in AD	[115]
**To confirm the relation between decreased BDNF and AD development**	Postmortem cortex	Low BDNF mRNA	RT-PCR	A decrease in brain-derived neurotrophic factor synthesis could significantly affect hippocampal, cortical, and basal forebrain cholinergic neurons and may account for their selective vulnerability in Alzheimer’s disease.	[116]
**Investigate plasma proteomic markers in early-onset versus late-onset AD**	Plasma BDNF	Elevated	Ultra-sensitive immuno-based assay	BDNF levels were elevated in both early-onset and late-onset AD	[117]
**Examination of BDNF serum level in elderly people**	Serum samples	No significant change	ELISA	There was no association between gender, depression, and dementia on serum level of BDNF.	[118]
**To assess BDNF serum and CSF concentrations in 30 patients at different stages of AD**	Serum, CSF	Early stages increased BDNF serum, decreased level in late stage	ELISA	BDNF can be a good determinant in the assessment of the progression of AD.	[119]

**Table 2 biomedicines-10-01143-t002:** Levels and effects of BDNF in Parkinson’s disease.

Study Objective	Sample Origin	BDNF Status	Assay Used	Conclusion	Ref.
**Investigating the effects of BDNF as a neuroprotective factor and as an adjunct therapy in PD**	NA	Decreased	NA	In animal PD models, physical activity increased the levels of BDNF and TrkB, which acted as a neuroprotective factor and resulted in symptomatic improvement	[133]
**Evaluate salivary cortisol and plasma BDNF levels in PD patients compared to healthy controls**	Plasma BDNF	No significant difference in BDNF, but higher cortisol in PD	ELISA	PD patients were in the early stage of the disease, so BDNF is not a suitable biomarker for early cases of PD	[134]
**Assess the association between neurotrophic changes and the clinical staging and motor severity of PD**	Peripheral BDNF	Decreased	ELISA	A larger decrease in BDNF (and other immune markers) were associated with a higher severity of PD	[135]
**Evaluate the levels of serum BDNF in recently diagnosed and untreated PD patients**	Serum BDNF	Decreased	Sandwich ELISA	Serum BDNF levels were lower in recently diagnosed and untreated PD patients compared to healthy controls	[136]
**To compare BDNF levels in PD, essential tremor (ET), and healthy controls**	Peripheral blood lymphocytes	Decreased in PD	Western blot	BDNF levels were decreased in PD patients, but no significant difference in ET patients	[137]
**Investigate the neuroprotective role of BDNF in PD mice**	NA	NA	NA	Elevating BDNF levels reduced mitochondrial impairment via increasing electron transport chain (ETC) activity and alleviating dopaminergic loss in PD mice	[138]

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
