# Peer review of "Brain-Derived Neurotropic Factor in Neurodegenerative Disorders"

_biomedicines, 2022, doi:10.3390/biomedicines10051143_

Round 1

Reviewer 1 Report

The authors reviewed BDNF in Alzheimer´s and Parkinson´s diseases. This is a relevant topic and has raised increasing research interest due to the epidemiology of neurodegenerative disorders. Nevertheless, the manuscript requires revision before being considered for publication.

Comments:

#1 – Since the authors did not include important acute disorders (ex. traumatic brain injury and stroke) or psychiatric disorders, the title of the revision is misleading.

I suggest that the title should be “BDNF in neurodegenerative disorders”.

# 2 – The title of Table 1 should be modified. I suggest “Levels of BDNF in Alzheimer-s disease”. There is no list of abbreviations as legend of the table. Furthermore, the studies reported in this table are former to 2010!

# 3 – A Table 2 with “Levels of BDNF in Parkinson´s disease” should be included.

# 4 - The authors mention that: … “this review aims to state the RECENT attempts relating the connection between the level of BDNF and the threat for Dementia and neurodegenerative disorders” ….

The most preoccupying aspect of this manuscript is the fact that a review should include recent literature, particularly when the objectives of the review state the authors are systematizing the recent attempts to relate BDNF and aging, AD as well as PD. These are topics effortlessly research nowadays. Extensive literature has been published regarding the crosstalk of BDNF and neurological disorders in the last 10 years. Thus, it is unacceptable that this review cited 131 references, but only 20 (15%) of the 131 references are from the past decade. Besides, of these 20 references (from 2012 -2022), 5 of them (25%) regard psychiatric conditions, that are not in the scope of this review!

It seems unconceivable a review in which 85% of the references cited are prior to 2012, particularly when the scope of the study regards BDNF and neurodegenerative conditions. Hence, I don´t have to mention that important studies, published recently, were not cited in the present manuscript.

Author Response

I am enclosing herewith revised review article entitled- "Brain-Derived Neurotropic Factor in Neurodegenerative Disorders(biomedicines-1616598) for consideration to be published in your journal of repute. We have in this review article emphasized on the role of the BDNF in the neurological diseases such as Parkinson's and Alzheimer.

Based on the comments from respected and learned editors and reviewers, we have added more information at appropriate places (highlighted in yellow). We have tried our best to address all queries raised. The comments are being replied and submitted in the authors response to reviewers file.

We are really thankful to the respected editor in chief, guest editors and learned reviewers for proving us with such an opportunity and seeding motivation for improvement of this article.

Reviewer 2 Report

Dear Authors,

The manuscript entitled "Brain-Derived Neutrotropic Factor in Neurological Disorders” is an interesting topic, however, there is a need for further scientific research on the topic. This is a review article and little current literature is addressed.

Some suggestions to scientifically improve the article.

- To review the recent literature, I suggest two articles, but there are many more.

- I suggest improving table 1, adding current information and creating a similar table 2 for Parkison's disease.

- There should be a greater contribution from the 10 authors of the article.

- Rewrite the keywords, there are too many and there should be no abbreviations

- Line 84 (The full stop should only come after the reference. Please rewrite)

- Line 114 (It makes two references to figure 1. Please rewrite)

- Was figure 1 created by the authors or adapted from an article?

- Line 116 (In legend, there are two types of punctuation “:.” Please rewrite.

- Line 160 - Pathological Mechanism of action, change to “Pathological Mechanism of Action”

- Table 1. (correct the legend formatting)

Table 1 should be reworded, there are much more recent articles on the topic. This article is a review article and as such should have a more up-to-date bibliography on the topic...

“Decreased Serum Brain-Derived Neurotrophic Factor (BDNF) Levels in Patients with Alzheimer’s Disease (AD): A Systematic Review and Meta-Analysis

Ted Kheng Siang Ng, Cyrus Su Hui Ho, Wilson Wai San Tam, Ee Heok Kua and Roger Chun-Man Ho”

“BDNF as a Promising Therapeutic Agent in Parkinson’s Disease

Ewelina Palasz, Adrianna Wysocka, Anna Gasiorowska, Malgorzata Chalimoniuk, Wiktor Niewiadomski and Grazyna Niewiadomska”

- Line 354 (Include the figure reference in the text and not alone.)

- Line 356 (In figure 2 legend, there are two types of punctuation “:.” Please rewrite.)

Author Response

(The authors gave the same response as above.)

Reviewer 3 Report

In the review Brain-Derived Neurotropic Factor in Neurological Disorders, the authors consider the role and mechanisms of action of BDNF in some pathological conditions.

Unfortunately, the purpose of this review is not clear. The authors do not write why their review is important and how it differs from numerous reviews published on this topic. The review does not provide a critical analysis of the current experimental data or highlight the key areas of research, etc.

It should be noted that the review cites mostly relatively old sources, for example, there are only 7 articles from 2017-2022 (5%), of which there are 5 review articles.

Although the title of the review suggests considering various neurological disorders, the article only considers 2 diseases: Alzheimer's disease and Parkinson's disease.

Again, I note that in Table "Studies on the BDNF and AD" only studies published prior to 2007 are cited. This is more than odd since there are two hundred articles published annually on this topic (PubMed).

Thus, this review is of little interest to readers.

Author Response

(The authors gave the same response as above.)

Round 2

Reviewer 1 Report

The author´s changed the title and included Table 1 and 2 listing studies of BDNF in AD and PD. Nevertheless, a great proportion of the references remain outdated. This is a point of major concern considering that the manuscript is a review of the literature.

Author Response

The references have been updated. Please see the highlighted changes.

Reviewer 2 Report

Dear authors,

There was an effort by the authors to improve the article, which added value to the article and to its publication in this journal. 
I would only suggest that the authors, at the end of the article, present the contribution of the various authors.

Author Response

(The authors gave the same response as above.)

Reviewer 3 Report

Unfortunately, I did not see that the authors substantially revised the review. My comment that the authors do not provide substantial analysis of recent works and new approaches still stands.

Author Response

(The authors gave the same response as above.)

Round 3

Reviewer 3 Report

Accept in present form